# Hyperuniformity and phase enrichment in vortex and rotor assemblies

Naomi Oppenheimer [1✉], David B. Stein [2], Matan Yah Ben Zion [3] & Michael J. Shelley [2,4✉]

Ensembles of particles rotating in a two-dimensional fluid can exhibit chaotic dynamics yet develop signatures of hidden order. Such rotors are found in the natural world spanning vastly disparate length scales — from the rotor proteins in cellular membranes to models of atmospheric dynamics. Here we show that an initially random distribution of either driven rotors in a viscous membrane, or ideal vortices with minute perturbations, spontaneously self assemble into a distinct arrangement. Despite arising from drastically different physics, these systems share a Hamiltonian structure that sets geometrical conservation laws resulting in prominent structural states. We find that the rotationally invariant interactions isotropically suppress long-wavelength fluctuations — a hallmark of a disordered hyperuniform material. With increasing area fraction, the system orders into a hexagonal lattice. In mixtures of two co-rotating populations, the stronger population will gain order from the other and both will become phase enriched. Finally, we show that classical 2D point vortex systems arise as exact limits of the experimentally accessible microscopic membrane rotors, yielding a new system through which to study topological defects.

---

[1] School of Physics and Astronomy, and the Center for Physics and Chemistry of Living Systems, Tel Aviv University, Tel Aviv 6997801, Israel. [2] Center for Computational Biology, Flatiron Institute, New York, NY 10010, USA. [3] Laboratoire Gulliver, UMR CNRS 7083, ESPCI Paris, PSL Research University, 75005 Paris, France. [4] Courant Institute, New York University, New York, NY 10012, USA. ✉email: naomiop@gmail.com; mshelley@flatironinstitute.org

Two-dimensional (or nearly so) fluid flows show rich and complex vortical dynamics. These can arise from flow interactions with boundaries[1,2], the inverse cascades of 2D turbulence[3-5], from Coriolis force-dominated atmospheric flows[6], and from quantization effects in superfluid He-II[7,8]. Point vortices have long been staples for the modeling of such inertially dominated inviscid flows. Kirchoff[9] was the first to describe point vortices using a Hamiltonian framework. His work was extended by many others[10-13], notably, Onsager[14] in his statistical mechanics treatment of 2D turbulence as clouds of point vortices.

Remarkably, structurally identical Hamiltonian and moment constraints can arise in the microscopic, viscously dominated realm from a strict balance of dissipation with drive on immersed rotating objects. These objects include models of interacting transmembrane ATP-synthase rotor proteins[15-17], and the planar interactions of rotors—microscopic particles driven to rotate by an external torque[18,19]. We refer to such systems as BDD systems, as in balanced drive and dissipation. In modeling rotational BDD systems, other physical effects may also come into play, such as steric interactions, that can yield interesting complexities[17]. Assemblies of interacting, driven-to-rotate particles have become an area of intensifying interest in the active matter community[18-26].

Here, we study both a BDD system of rotating microscopic particles—membrane rotors—immersed in a flat membrane, and point vortices which are a particular limit of this BDD system. For both, symmetries in the Hamiltonian, $\mathcal{H}$, lead to conservation laws that are geometrical in nature, bounding the proximity and distribution of particles. We derive a connection between the Hamiltonian and the structure factor, $S(\mathbf{q})$ (where $\mathbf{q}$ is the wavevector), which can be used to place bounds on spatial correlations,

$$\mathcal{H}[\rho(\mathbf{r})] = \frac{N\Gamma^2}{4\pi} \int \mathbf{dq} S(\mathbf{q}) \widetilde{\Psi}(\mathbf{q}), \quad (1)$$

where $\widetilde{\Psi}$ is the Fourier transform of the stream function. In the case of point vortices, $\widetilde{\Psi}(\mathbf{q}) = 1/q^2$, where $q = |\mathbf{q}|$. As we show, Eq. (1) argues that this system should tend towards hyperuniformity. That is, the long-wavelength configuration at steady state is characterized by an isotropically vanishing structure factor, $S(\mathbf{q} \to 0) \to 0$, leading to an isotropic band-gap[27-29]. To investigate this prediction, we numerically simulate assemblies of both BDD and point vortices and observe (see Fig. 1): (i) hyperuniformity for BDD systems; (ii) evidence that point vortex systems can become hyperuniform depending on how they are perturbed; (iii) phase enrichment (in both cases); and (iv) crystallization (for BDD). Our observations lead us to conclude that rotational dynamics provide a mechanism for the self-assembly of particles into a disordered hyperuniform 2D material.

## Results

**Dynamics of vortex/rotor ensembles.** We begin by introducing a single vortex in an ideal inviscid fluid. We then describe the flow generated by a point rotor in a viscous membrane and show that the two flows are identical in a biologically relevant limit. We use this equivalence and apply known tools from the study of ideal vortices on both systems. Namely, the linearity of the equations enables extending the result of a single vortex to the flow generated by an ensemble of vortices, which, in turn, could also be described by a Hamiltonian.

An ideal point vortex is given by a singular vorticity, $\boldsymbol{\omega} = \nabla \times \mathbf{v} = \delta(\mathbf{r})$. A 2D incompressible fluid can be described using a stream function $\Psi$ such that the velocity, $\mathbf{v}$, is given by $\mathbf{v} = \partial^\perp \Psi$. This equation, combined with the equation above gives, $\Psi = -\frac{1}{2\pi} \log r$ (ref. [12]). The flow, $\mathbf{v}(\mathbf{r})$, therefore, scales as $1/r$, where $r = |\mathbf{r}|$.

We switch now to a point rotor in a viscous membrane, driven by an external torque $\tau$ (see Fig. 2A for a schematic representation). Following Saffman and Delbrück's seminal work[30], and others that followed[15,16,31-33], we assume that the membrane is incompressible ($\nabla \cdot \mathbf{v} = 0$), and that inertia is negligible. Under these assumptions, the Stokes momentum conservation equation for the membrane reads,

$$0 = \eta_{2D}\nabla^2\mathbf{v} + \eta_{3D}\frac{\partial \mathbf{u}^\pm}{\partial z}\bigg|_{z=0^\pm} + \tau\partial^\perp\delta(\mathbf{r}), \quad (2)$$

where $\mathbf{v}$ is the 2D velocity in the plane of the membrane, $\mathbf{u}^\pm$ is the 3D flow in the outer fluids, $\eta_{2D}$ is the 2D viscosity, and $\eta_{3D}$ is the viscosity of the outer fluids. The second term on the right-hand side is the surface shear stress of the outer fluids, and the third term is the force due to a rotating point object. There is no pressure contribution when the motion is purely rotational. This equation is coupled to the equations of the outer fluids. It is easy to solve the above equations using a 2D Fourier transform ($\widetilde{F}(\mathbf{q}) = \int_{-\infty}^{\infty}\int_{-\infty}^{\infty}F(\mathbf{r})e^{-i\mathbf{q}\cdot\mathbf{r}}d^2r$), giving:

$$\widetilde{\mathbf{v}}(\mathbf{q}) = \Gamma\partial^\perp\widetilde{\Psi} \;;\; \widetilde{\Psi} = \frac{1}{q(q+\lambda^{-1})}, \quad (3)$$

where $\Gamma = \tau/\eta_{2D}$, and $\lambda = \eta_{2D}/2\eta_{3D}$ is the Saffman Delbrück length. At small distances ($r \ll \lambda$), momentum travels in the plane of the membrane. At large distances ($r \gg \lambda$), momentum travels through the outer fluid as well[34,35]. In real space, $\Psi(\mathbf{r}) = 1/4(H_0(r/\lambda) - Y_0(r/\lambda))$, where $H_0$ and $Y_0$ are zeroth-order Struve function and Bessel function of the second kind, respectively.

In the limit of small distances, $r \ll \lambda$, the stream function is, $\Psi \approx -\frac{1}{2\pi}\log r$, i.e., exactly the same as for an ideal point vortex. In the opposite limit, $r \gg \lambda$, the stream function becomes $\Psi = \frac{1}{2\pi r}$ as in quasigeostrophic (QG) flows—atmospheric or oceanic flows coming from gradients in pressure coupled to the Coriolis force[36],

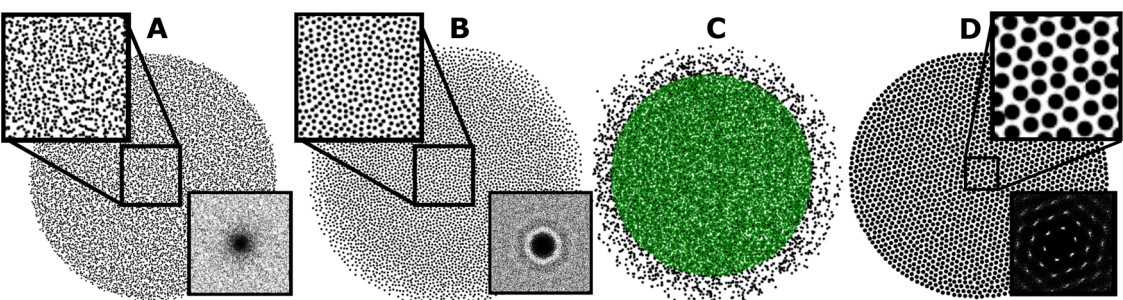

**Fig. 1 Three different structural states of 2D vortices/rotors.** (**A**) Hyperuniformity for Euler point vortices, (**B**) Hyperuniformity for QG rotors/surface rotors, (**C**) Phase enrichment induced by circulation differences where green (black) represents vortices of high (low) circulation, and (**D**) Crystallization arising from hydrosteric interactions. The insets of (**A**), (**B**), and (**C**) show the structure factor, $S(\mathbf{q})$. In (**A**) and (**B**), $S(q)$ decays to zero at small $q$, indicating that the distribution is hyperuniform. In (**C**), the structure factor shows the six distinct peaks of a hexagonal lattice.

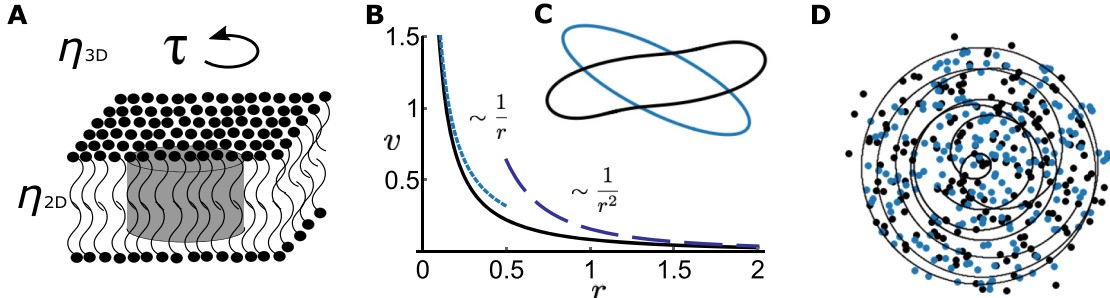

**Fig. 2 Fluid flow and dynamics for membrane rotors. A** A representation of a membrane rotor—a disk rotating due to a torque $\tau$ in the plane of the membrane. **B** The velocity field due to a membrane rotor (solid line) which scales as a point vortex $v \sim 1/r$ at small distances (dotted), $r/\lambda \ll 1$, transitioning to a QG behavior at large distances $v \sim 1/r^2$ (dashed). **C** Contour dynamics of an ellipse with radii ratios $r_l/r_s \leq 3$, where $r_l$ ($r_s$) is the major (minor) axis. Starting from the same contour, the dynamics differ according to the radius relative to the SD length. Blue is in the limit $r_l \ll \lambda$. In this limit, the ellipse is rotating as a rigid body, as predicted by Kelvin[65] for an elliptic patch in an Euler fluid. Black is in the limit $r_l \gg \lambda$, no longer conserving its shape since the large distance flow is in the quasigeostrophic regime. **D** 200 point membrane rotors, blue is the initial random configuration, black is the final configuration. Solid line shows typical trajectory of an individual vortex. Note that the area did not change considerably since the system of vortices is self-bounding.

or driven rotors on the surface of a fluid[22]. A membrane rotor, therefore, transitions from a point vortex for Euler at small distances to that of QG flow at large distances. Thus, the velocity diverges (decays) as $1/r$ ($1/r^2$) in the limit of small (large) distances (see Fig. 2B, C). For simplicity, we work primarily in the limit of small distances, $r \ll \lambda$, since in this limit the dynamics in a membrane converge with those of point vortices (many results still apply to the more general case as shown in the Supplementary Figs. 5 and 6). In what follows, we will use the term point vortices when there are only hydrodynamic interactions and the term rotors when the particles have steric interactions in addition to hydrodynamic ones.

The dynamics of $N$ point vortices are dictated by Hamilton's equations,

$$\Gamma_i \mathbf{v_i} = \partial_i^\perp \mathcal{H}, \quad \text{with} \quad \mathcal{H} = \frac{1}{2}\sum_{i \neq j} \Gamma_i \Gamma_j \Psi(|\mathbf{r}_i - \mathbf{r}_j|), \quad (4)$$

where $\partial_i^\perp = (\partial y_i, -\partial x_i)$, $\mathbf{v_i}$ is the velocity of the $i$th vortex, and $\Gamma_i$ is the circulation (proportional to the magnitude of the torque for rotors, $\Gamma_i = \tau_i/\eta_{2D}$). The Hamiltonian depends on the conjugate variables $\mathbf{r}_i = (x_i, y_i)$, [normalized by the circulation $\sqrt{|\Gamma_i|} \, \text{sgn}(\Gamma_i)$], i.e., the positions of the vortices[12]. The symmetries of the Hamiltonian correspond to conservation laws[37]. In this case, we have symmetries with respect to translation in time, space, and rotation, corresponding to the conservation of the Hamiltonian itself, and of the first and second moments of vorticity, $\mathbf{L} = \sum_i \Gamma_i \mathbf{r}_i = \mathbf{0}$ wlog, and $M = \sum_{i,j} \Gamma_i r_i^2$. The conservation of $\mathbf{L}$ and $M$ are analogous to the conservation of the center of mass and to the moment of inertia, with sums weighted by circulation instead of mass. From the conservation laws we can deduce that the initial area cannot change dramatically. Particles cannot drift to infinity since the second moment is fixed, nor can they collapse to a point since the Hamiltonian is conserved. These properties are readily observed in simulations. Figure 2D shows typical trajectories of 200 membrane rotors. The initial distribution is random in a predefined finite area, and the dynamics are chaotic[38]. The final configuration occupies nearly the same region of space as the initial configuration does, and the conservation laws hold to high precision in our simulations, as detailed in "Methods". This self confining property of vortex dynamics has further consequences, as we now show.

**Hyperuniformity.** Hyperuniformity is the suppression of density-density fluctuations at small wavenumbers (or correspondingly, at large distances)[39–41]. Disordered hyperuniformity can emerge due to

short-ranged interactions such as those that arise in sheared suspensions[42–44], jammed materials[45], and for spinning particles[46]. Here, we will show hyperuniformity emerging from long-ranged interactions, similar to its emergence in sedimentation of irregular objects[47]. A good way to characterize hyperuniformity is the structure factor, defined as $S(\mathbf{q}) = N^{-1}|\widetilde{\rho}(\mathbf{q})|^2$, where $\rho(\mathbf{r}) = \sum_i \delta(\mathbf{r} - \mathbf{r}_i)$ is the coarse-grained density. In a hyperuniform material, $S(q)$ goes to zero as a power law at small wavenumbers. We present an argument that a density of point vortices should be hyperuniform due to the conservation of the Hamiltonian. For a density of rotors, the Hamiltonian is given by $\mathcal{H}[\rho(\mathbf{r})] \sim \frac{\Gamma^2}{2}\int d\mathbf{r}\int d\mathbf{r}'\rho(\mathbf{r})\rho(\mathbf{r}')\Psi(|\mathbf{r} - \mathbf{r}'|)$. Using the convolution theorem, we find a general relation between the Hamiltonian and the structure factor given by Eq. (1). In the case of point vortices, $\widetilde{\Psi}(\mathbf{q}) = 1/q^2$, which gives

$$\mathcal{H}[\rho(\mathbf{r})] = \frac{N\Gamma^2}{2}\int \frac{S(\mathbf{q})}{q}\,dq. \quad (5)$$

For the integral of Eq. (5) to converge in 2D, $S(\mathbf{q}) \sim q^\alpha$ near the origin, and we must have $\alpha > 0$. In other words, an ensemble of point vortices should be hyperuniform. Figure 3B shows an apparent $\alpha \sim 1.3$ scaling for point vortices, consistent with the above argument.

Using simulations, we show that a set of $N$ vortices, uniformly distributed within a radius $R$, evolves to a disordered steady state with a hidden order visible to the naked eye (compare Fig. 3A left and right). We quantitatively characterize the system in steady state in three ways: (1) *The structure factor*: At steady-state $S(\mathbf{q})$ shows a distinct cavity, at $q \approx 0$, $S(\mathbf{q}) \to 0$, for both points vortices (Fig. 3A) and rotors (Fig. 3C). (2) *Perturbations*: We demonstrate that hyperuniformity is robust under different perturbations, be it in the form of numerical errors, repulsive interactions, or impurities (in the next section). For point vortices, the steady state appears later and later as the timestep is decreased (see Supplementary Fig. 1), suggesting that perturbations are necessary for convergence, here very small but persistent time-stepping errors[48]. We suspect that perturbations that break the rotational symmetry of the Hamiltonian are required, as testing a smaller ensemble of 1000 point vortices with a symplectic integration scheme, based on the exact solution of pair interactions[49], showed little sign of hyperuniformity (see Supplementary Fig. 2 and "Methods"). The observed relaxation toward hyperuniformity is consistent with the critical slowing down reported for other systems[40]. Adding steric interactions, hyperuniformity appears on a timescale that is independent of the timestep. Moreover, with steric interactions, as the area fraction $\phi$ of the particles is increased, the system transitions

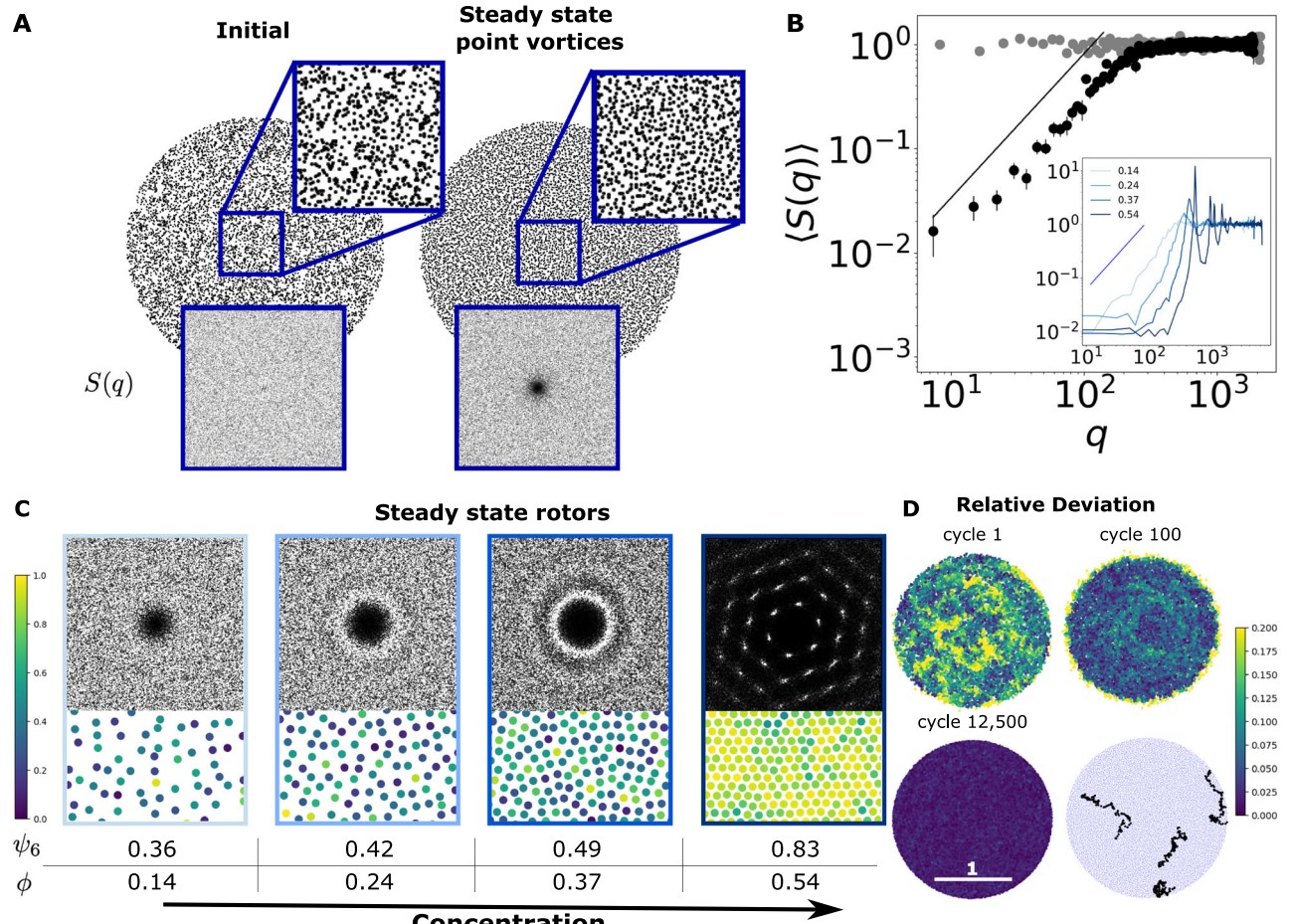

**Fig. 3 Hyperuniformity in ensembles of point vortices and rotors. A** Snapshots of 10,000 point vortices initially (left) and at steady state (right). Insets show the structure factor, $S(\mathbf{q})$ with a distinct cavity at steady state. **B** Angular average of the structure factor shown in (**A**), in a log-log scale with solid line showing a $q^{1.3}$ scaling. Error bars are standard deviation over ten well-separated timesteps. Inset shows the structure factor of the rotors shown in (**C**) with increasing hue corresponding to increased concentration $\phi = (0.14, 0.24, 0.37, 0.54)$. Solid line is the same $\alpha \sim 1.3$ scaling. **C** Steady-state configurations of 2000 membrane rotors with the corresponding structure factors, showing a transition from disordered hyperuniformity to a hexagonal lattice. Particles are colored according to their local bond orientation parameter $\psi_6$. For particle $j$, $\psi_6^j = \sum_i e^{6i\theta_{ij}}$, with the sum taken over nearest neighbors as found by a Voronoi diagram. The table gives ensemble-averaged values, $N^{-1}\sum_j \psi_6^j$. **D** A plot of the relative deviation for each particle, with the relative deviation of particle $i$ defined by how far it is displaced from its position at the previous cycle, i.e., $|r_i(t + t_{cyc}) - r_i(t)|$. The cycle time is calculated at steady state as the average time it takes the system to rotate by $2\pi$. Particles are colored by their relative deviation, from blue to yellow with increasing deviation. The plot at the bottom shows the strobed position of four particles during a time interval of $\Delta t \approx 115$ cycles; the particles in the strobed frame move along Brownian-like trajectories.

from disordered hyperuniform, to an ordered hyperuniform hexagonal lattice at $\phi \sim 0.5$, as can be seen in Fig. 3C (as a sanity check we show in Supplementary Fig. 3 that a confined, rotationally sheared suspension does not become hyperuniform). The inset of Fig. 3B shows the averaged structure factor where at intermediate area fractions we see Percus–Yevick type features for the structure factor of disks[50]. (3) *The relative deviation:* We observe that at late times the ensemble of point vortices rotates almost as a rigid body and each particle nearly goes back to its position at the previous cycle (see Supplementary Fig. 4). The system may seem to have reached an absorbing state, but note that the relative deviation (as defined in Fig. 3D) only measures changes over a single cycle. The motion of vortices over many cycles is still chaotic. Figure 3D shows the trajectory of a few vortices over ~115 cycles, showing Brownian-like dynamics. Similar results were obtained for membrane rotors at an area fraction of $\phi = 0.1$. For sufficiently large area fractions, the system crystallizes and the ensemble rotates as a rigid body where relative deviations are close to zero over many cycles.

**Rotation-induced phase enrichment.** We now show that for mixed populations of fast and slow rotating particles, there is phase enrichment of both populations and hyperuniformity of the fast ones. Consider a mixture of two equally numbered populations ($\rho_l = \rho_h$ at $t = 0$) initially placed within the same radius R. $\rho_l$ rotates slowly with $\Gamma_l \ll \Gamma_h$, where $\Gamma_h$ is the circulation of the second population. Figure 4A shows long-time simulation results for 10,000 point vortices. The two populations behave very differently. The fast vortices remain in a disk of only slightly smaller size than their initial area (Fig. 4B). The slow particle distribution shows a significant expansion. In addition, there is a striking difference when comparing the independently computed structure factors of these two populations, the fast vortices are hyperuniform with $S(q) \sim q^{1.4}$, whereas the slow ones show no signs of hyperuniformity (Fig. 4C). This difference is dramatic enough to be visible in a cursory examination of the separate distributions; see Fig. 4A.

A heuristic model sheds light on this phenomenon (see Supplementary Note 1). Above, each vortex population starts

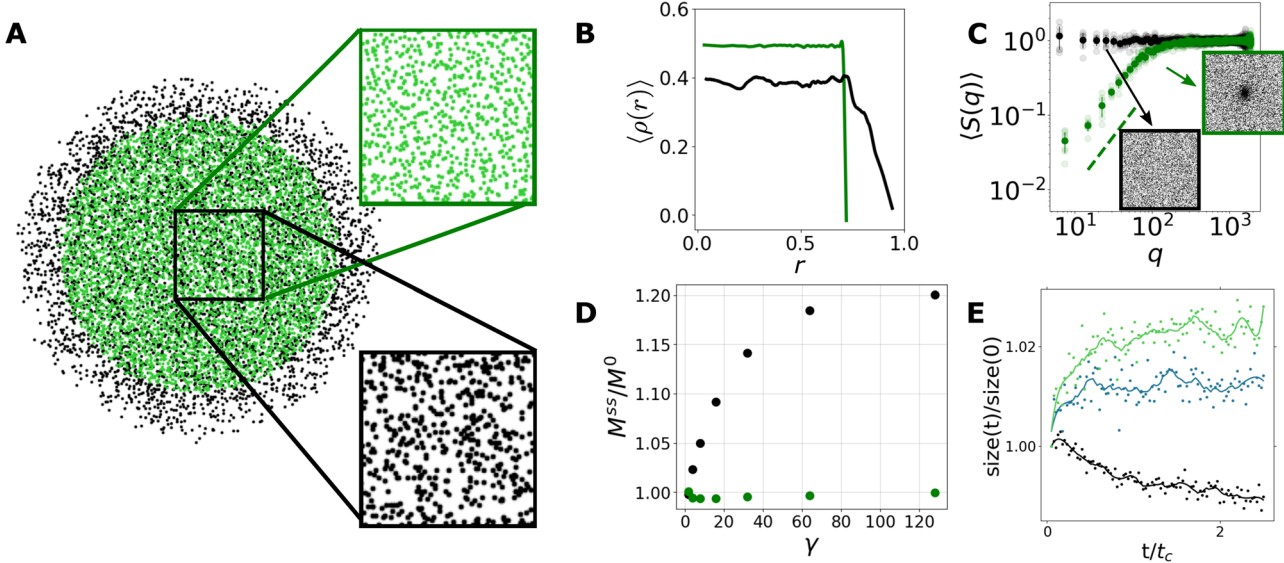

**Fig. 4 Two populations of vortices with different circulations showing phase enrichment,** $\Gamma_l = 2\pi$ **in black and** $\Gamma_h = 256\pi$ **in green. A** Steady-state configuration for ten thousand point vortices of a circulation ratio $\gamma = \Gamma_h/\Gamma_l = 128$. Each inset shows a close-up view of one of the populations within the same physical region. **B** Density of the configuration in (**A**), $\rho(r)$, averaged over angle as a function of distance from the center. Note how density fluctuations are suppressed for the high-circulation vortices, as is more clearly observed by the averaged structure factor, $S(q)$, in (**C**), where the dashed green line shows a $\sim q^{1.4}$ power law. Error bars are standard deviation over ten well-separated timesteps shown as transparent dots on top of the average result. **D** The second moment for $N = 10,000$ vortices. Plotted separately for the high (in green) and low (in black) vortices at steady state as a function of $\gamma$ (i.e., increasing $\Gamma_h$). **E** LOSSLESS compression for the two populations showing an increase (decrease) in file size (an estimate of entropy) for the low (high) circulation vortices over a couple of cycles. In blue is the file size for the total system. Solid line is a moving average, time is normalized by an average cycle time $t_c$.

with uniform density within the disk of radius $R$. Consider as a candidate long-time solution one where each population remains of uniform density, $\rho_l$ and $\rho_h$ respectively, and confined within concentric circles of radii $R_l$ and $R_h$. The circulations of each population, $\Gamma_l$ and $\Gamma_h$, and the system Hamiltonian $H$ and second moment $M$ are fixed by the initial configuration, which restricts the possible values of $R_l$ and $R_h$. There are two possible solutions. In the first, $R_l = R_h = R$. In the second, the radius of the fast vortices slightly decreases to $R_h$, allowing the slow vortices to expand to a larger radius $R_l$ given by $R_l^2 = (\gamma + 1)R^2 - R_h^2\gamma$, where $\gamma = \Gamma_h/\Gamma_l$ (see Fig. 4D). For large $\gamma$, we find that $R_h \simeq R(1 - \beta/\gamma)$, where $\beta$ is a positive prefactor of order 1. The slow vortices radius asymptotes to $R_l = R\sqrt{1 + 2\beta} + O(1/\gamma)$. The numerical results indicate that the outer radius indeed asymptotes to a finite constant as $\gamma \to \infty$ (see Fig. 4D and Supplementary Figs. 6–8).

Why are solutions with two different radii those observed in our simulations? Such a solution is favored entropically as it maximizes the number of available states. At large $\gamma$, the main entropical contribution is volumetric, $\Delta\mathcal{S}_{\text{volume}} = 2N\log(R_{\text{final}}/R_{\text{initial}})$. Since the high-circulation vortices hardly change radius, $R_h \xrightarrow{\gamma\to\infty} R$, the change in entropy is coming mainly from the expansion of the low circulation vortices and is given by $\Delta\mathcal{S}_{\text{total}} \sim N\log(1 + 2\beta) > 0$. Coupling the two populations allows one population to expand where before it was bounded. The situation is analogous to depletion interactions, where the net entropy of a system increases by condensing the large particles allowing for the small particles to explore a larger volume[51]. As Onsager first suggested[14], in a bound system, configurational entropy must have a maximum as a function of energy above which demixing of two populations can be observed.

A simple way to estimate the entropy in a system is by using LOSSLESS compression, as suggested by refs. [52,53]. Compressing plots of particle positions in a system of 10,000 point vortices with

circulation ratio $\Gamma_h/\Gamma_l = 128$ shows an increase in file size for $\rho_l$ and a decrease for $\rho_h$, while the combined system is increasing, see Fig. 4E.

## Discussion

We have shown that driven rotors in a membrane or a soap film, like point vortices in an ideal 2D fluid, have geometrical conservation laws which limit their distribution. These conservation laws suggest different possible structural states such as hyperuniformity and phase enrichment. We suspect that a completely pure system of point vortices may never reach hyperuniformity due to a dynamical bottleneck, but have shown that hyperuniformity is robust to two forms of perturbations, whether arising due to numerical errors or steric interactions. For rotors with steric interactions, the unbounded ensemble crystallizes into a hexagonal lattice when the area fraction $\phi \gtrsim 0.5$ (see also ref. [17]). We have limited the discussion to membrane rotors and vortices, but the results for hyperuniformity and phase enrichment hold for other settings in which particles are restricted to a 2D plane, e.g., rotors at the surface of a fluid (see Supplementary Figs. 5 and 6). In fact, while this paper was under review, hyperuniformity was reported in populations of algae swimming in right circles at an interface[54].

What is especially interesting about our particular BDD system is its potential for experimental realizability, its moment and Hamiltonian structure, and that its near-field interactions (i.e., below the Saffman–Delbruck length) are identical to those of Euler point vortices. Further, the far-field interactions of membrane rotors are identical to those of point vortices of the semi-quasigeostrophic equations[36,55,56] used to model atmospheric flows. Thus, to observe the interesting dynamical features we describe, one does not need to go to the atmospheric scale, or cool a fluid to near-zero temperature. In principle, one can simply observe microscopic particles on a soap film, in smectic films, a membrane, or even at the surface of a fluid[19,22,57,58].

## Methods

**Simulations.** Simulations were performed in Python. Random initial configurations within the unit disk were found by rejection sampling (points in the unit rectangle were sampled uniformly, transformed to the rectangle $[-1, 1]^2$, and those whose radius exceeded the target radius were discarded). The initial Hamiltonian $H_0$ and second moment $M_0$ are computed at $t = 0$, and the relative errors $\epsilon_H(t) = |H_t - H_0|/H_0$ and $\epsilon_M(t) = |M_t - M_0|/M_0$ are monitored as a measure of fidelity. For simulations of rotors (i.e., with steric repulsion), a 5th order explicit Runge–Kutta method based on the Dormand–Prince scheme[59] with a fixed timestep size of $\delta t = 10^{-7}$ was used. Long integration times were required for simulations of point vortices, and for these simulations an explicit eighth-order adaptive method based on the Dormand–Prince scheme[60,61] was used, with both relative and absolute tolerances set to $10^{-6}$. The specific implementation of the scheme used was the *DOP853* method of *scipy.integrate*[62]. For simulations of 10,000 point vortices with $\Gamma = 2\pi$, $\epsilon_H(t) < 1.5 \times 10^{-3}$ and $\epsilon_M(t) < 4 \times 10^{-5}$ up to $t \approx 1.6 \times 10^4$ cycles, while for simulations with 5000 vortices with $\Gamma = 2\pi$ 5000 vortices with $\Gamma = 256\pi$, $\epsilon_H(t) < 0.1$, and $\epsilon_M(t) < 4 \times 10^{-5}$ up to $t \approx 10^5$ cycles. Time is normalized by the average cycle time, $t_c \approx 4\pi^2 R^2/\sum_i \Gamma_i$, where $R$ is the initial radius. We tested running these simulations for 1000 particles with symplectic time integration based on an exact solution of two point vortices[49]. Simulations were run up to $t \sim 1.4 \times 10^5$ cycles. Due to numerical constraints, we did not run larger ensembles or longer times. At these times, we did not observe clear signatures of hyperuniformity, though there was an indication of a slight decrease in $S(q)$ for low $q$. Beginning the simulation with a hyperuniform state, and running the symplectic integration over $5 \times 10^4$ cycles preserved hyperuniformity (see Supplementary Fig. 1).

Steric interactions were taken as the repulsive part of a harmonic potential, i.e., for two particles whose centers are distance $r_i$ apart, $F = -kr_{ij}$ if $r_{ij} < 2a$ and zero otherwise. The use of a harmonic potential, rather than a sharp step function for hard core particles, provided improved numerical stability and convergence. A large $k$ value was chosen to ensure no overlap between particles, $k = 10^6$, for particles of radius $a = 0.01$.

**Structure factor.** To accurately compute the structure factor $S(\mathbf{q})$ we use a type-1 non-uniform fast-Fourier transform[63]. Explicitly, points are restricted to a windowing region that is confined entirely within the unit disk. The frequencies $\widetilde{\rho}(\mathbf{q})$ are computed for the first 512 modes in each direction, and the average value (i.e., $\widetilde{\rho}(0)$) is set to 0. This results in structure factors in the plane, such as those shown in Fig. 3. Except in those cases where crystallization occurs, these structure factors are azimuthally isotropic. To summarize this information, the angular average over the structure factor was calculated by slicing the result to 1000 equal bins between $q_{min}$ and $q_{max}$ and taking the mean of the results that fell within each slice.

**Compression.** A plot of the positions of the point vortices was compressed using PNG with AGG backend. Each vortex was plotted by a single pixel. The total size of the plots was kept fixed in time. The figure size was chosen to minimize overlap between neighboring vortices but maintaining a computationally accessible file size.

## Data availability

All data that support the findings of this study can be reproduced using the code available on https://github.com/dbstein/rotor_hyperuniformity[64]. Data files are also available upon request.

## Code availability

The code for generating and analyzing the data presented in this paper is available at https://github.com/dbstein/rotor_hyperuniformity[64].

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

## Acknowledgements

We thank Haim Diamant for insightful discussions regarding the emergence of hyper-uniformity from the conservation laws, to Martin Lenz for suggesting a simple heuristic model of the phase enrichment, and to Enkeleida Lushi. N.O. acknowledges support by the Israel Science Foundation (grant No. 1752/20). M.J.S. acknowledges support by the National Science Foundation under Awards Nos. DMR-1420073 (NYU MRSEC), DMS-1620331, and DMR-2004469.

## Author contributions

M.J.S. and N.O. initiated the research. N.O., D.B.S., and M.Y.B.Z. wrote initial versions of the code. Final versions of the code were written by D.B.S.. N.O. and D.B.S. analyzed the data. N.O. developed theory. All authors conceptualized, designed, reviewed the work, suggested ideas, and wrote the manuscript.

## Competing interests

The authors declare no competing interests.
