## [Peer Review File · Nature Communications]

Hyperuniformity and phase enrichment in vortex and rotor assembliesREVIEWER COMMENTS

Reviewer #2 (Remarks to the Author):

The paper describes simulation results for two related systems of 2D interacting vortices: 1. Simplified point vortices and 2. More realistic rotor assemblies with soft-core steric repulsion and long-range Saffman-Delbruck interaction. The main results are that both systems show hyperuniformity at a range of parameters. In particular, hyperuniformity does not require fine tuning.

This is a well-written, interesting and original paper, of clear interest to researchers in a range of fields, including random-media, Hamiltonian systems, dynamics in membranes and more. Realistic examples demonstrating hyperuniformity are few, thus, the paper is a significant contribution to the field. Accordingly, I recommend the paper is accepted to Nature Communication pending revision and some clarifications as detailed below.

My main concern is regarding the numerical method used to perform the simulation. In particular, I do not understand the dependence of the rate of convergence to the steady state on the step size and the relevance of ref (49). As far as I understand, in (49), the role of the added noise is to break the rotational symmetry in the numerical solution. In contrast, here, the authors claim that conservation of H and M is essential. What happens to fig 2E at long times with added noise? Fig 2E (which can move to the SM) should show the error in H and M for the entire duration of the simulation, not just a few cycles. If the errors (including noise) remain negligible, it may not be necessary to re-do the simulations. However, if errors grow, this may indicate numerical issues. Given the Hamiltonian structure, which is emphasized by the authors but ignored in choosing the appropriate numerical method, it makes more sense to use a symplectic method.

Currently, the results do not exclude the possibility that the hyperuniform steady-state is a numerical artifact. In other words, while a hyperuniform steady state may exist, a dynamical bottleneck may not allow the conservative system to reach it. In this case, initial conditions may also be a factor. To summarize, the authors need to explore the impact of the applied numerical methods on the results and establish their validity.

Minor comments

1. In the subplots showing the 2D $S(q)$, I find the color scheme (black = zero) confusing. I recommend it is inverted.
2. Page 2, 11 lines from the end: typo – extra `(`.
3. What happens at low densities? Is the steady state still hyperuniform?
4. The definition of returnity can be clarified.
5. Entropy measurements and Fig 4E: The changes in entropy are very small – about 1-2% with an error of 0.5-1%. This is not surprising given that converting the data into a figure with $K \times K$ pixels, estimating the entropy using lossless compression will be severely undersampled unless the number of realizations is of the order of $2^{(K^2)}$. As the authors already calculated the structure factor, I suggest applying the method of [43]. Also, it is interesting to plot the entropy as a function of Γ_h/Γ_l .

Reviewer #3 (Remarks to the Author):

This manuscript by Oppenheimer et al. "Hyperuniformity and phase enrichment in vortex and rotor assemblies" employs theory and simulations to describe the dynamics of point vortices and finite size rotors in 2-dimensions. The authors show that the 2D rotational flows can lead to ordered states, where hyperuniformity, phase enrichment or hexagonal order can be observed.

By considering Hamiltonian formalism, the authors predict that point vortices should form disordered hyperuniform states due to conservation laws. The authors also recover the exactly the same result for torque driven membrane rotors with a finite size, in the limit of short separations. The theoretical calculations are supported by detailed hydrodynamic simulations, with an excellent agreement between the two. Simulations are then used to consider a 50:50 mixtures of fast and slow spinning point vortices. Here, phase enrichment is observed for both population, while hyperuniformity is observed for fast spinning vortices only. The two populations occupy two different areas, which is shown to maximize the number of available states.

From the point of view of rotational active matter, I do find the presented results interesting. It is fascinating that the conservation laws stemming from the Hamiltonian structure of the rotationally invariant interactions result to the distinct states, as shown by the authors. Especially the phase enrichment in the mixtures is very nice. I am not a theorist nor too familiar with the formalism used, so some details may have escaped from me. The authors make an effort to simplify the mathematics, but at places I was lost what were the basic assumptions vs derived results.

Generally, the results are interesting, nicely presented, and most parts are clear. The article probably deserves publication, but I would encourage the authors to clarify the general presentation/arguments, clearly state what is assumed and what are the corresponding results, and give intuitive explanations if possible. These would render the article more accessible.

Other comments/questions:

- 1) On the page 3, where the steady state is characterized by 3-distinct ways. The role of perturbations is discussed. It is stated that some amount of perturbations are needed to observe hyperuniformity. While this would not be a problem in an experimental setting, does it imply that if the Hamiltonian and its moments are exactly conserved, hyperuniformity would not be observed for point vortices?
- 2) The $S(q)$ for rotors (inset in Fig. 3B) shows plateau at small q of all $\phi > 0.14$. While in the main text it is said that all the samples showed hyperuniformity. This seems contradictory with the definition of $S(q) \rightarrow 0$ when $q \rightarrow 0$. Also, the rotors probably develop (local) hexatic order before the crystallization observed at $\phi \sim 0.5$. It would instructive to plot the local hexatic order parameter as a function of ϕ .
- 3) I was somewhat confused with the returnity. The returnity seems to be close to zero at longtimes. Logically to me this suggests that the particle position has deviated from its original position. However, the definition of the returnity in the caption suggests that zero returnity means no deviation. The calculation/discussion should be clarified.
- 4) Connected to above, from the main text it seems that the system is not in absorbing state at long times, while it is also said that the ensemble undergoes a solid body rotation, where the particle i returns to its position cyclically. This paragraph should be clarified.
- 5) For which system is the returnity measured? Would it be expected to be different between point vortices and rotors?
- 6) The role of fig. 2c is not clear to me. Does this mean that at long separations (low area/volume fractions) the finite size rotors are not expected to have similar steady state as predicted for point

vortices?

7) Also, if not completely obvious, it would be nice if arguments/simple explanations were given for example for the symmetries for a translation in time, space and rotation corresponding to the conservation laws. This would increase the accessibility of the article. Can these be understood in some simple way, such as conservation of the hamiltonian and 2nd moment kind of fixes the density, while the vanishing first moment corresponds to no center of mass movement? At longtimes this then leads to a steady state with a (close to) solid body rotation, as opposite to the chaotic dynamics at times near the random starting configuration.

8) In the ref 15, for torque driven rotors it is suggested that the steady state is characterized by extrema of a "pseudo kinetic energy". In the formalism presented here, is there an analog to this?

9) The ψ in the Hamiltonian above the equation 5 on page 3 is not defined.

Below we copy the referee reports in green. Our response is in black below each point.

Report and response to Referee 2:

Referee: The paper describes simulation results for two related systems of 2D interacting vortices: 1. Simplified point vortices and 2. More realistic rotor assemblies with soft-core steric repulsion and long-range Saffman-Delbruck interaction. The main results are that both systems show hyperuniformity at a range of parameters. In particular, hyperuniformity does not require fine tuning.

This is a well-written, interesting and original paper, of clear interest to researchers in a range of fields, including random-media, Hamiltonian systems, dynamics in membranes and more. Realistic examples demonstrating hyperuniformity are few, thus, the paper is a significant contribution to the field. Accordingly, I recommend the paper is accepted to Nature Communication pending revision and some clarifications as detailed below.

Response: We thank the referee for finding our paper well-written, interesting and original! We further appreciate the pertinent questions raised which we hope we properly address in our response below as well as in the revised manuscript. By trying to address the reviewer’s concern we feel that we have not only improved the paper, but also our understanding of the system and its dynamics.

Referee: My main concern is regarding the numerical method used to perform the simulation. In particular, I do not understand the dependence of the rate of convergence to the steady state on the step size and the relevance of ref (49). As far as I understand, in (49), the role of the added noise is to break the rotational symmetry in the numerical solution. In contrast, here, the authors claim that conservation of H and M is essential. What happens to fig 2E at long times with added noise? Fig 2E (which can move to the SM) should show the error in H and M for the entire duration of the simulation, not just a few cycles. If the errors (including noise) remain negligible, it may not be necessary to re-do the simulations. However, if errors grow, this may indicate numerical issues. Given the Hamiltonian structure, which is emphasized by the authors but ignored in choosing the appropriate numerical method, it makes more sense to use a symplectic method.

Currently, the results do not exclude the possibility that the hyperuniform steady-state is a numerical artifact. In other words, while a hyperuniform steady state may exist, a dynamical bottleneck may not allow the conservative system to reach it. In this case, initial conditions may also be a factor. To summarize, the authors need to explore the impact of the applied numerical methods on the results and establish their validity.

Response: The referee raises a valid concern. In fact, we believe that the referee is exactly correct when he states that “while a hyperuniform steady state may exist, a dynamical bottleneck may not allow the conservative system to reach it”. We had written something to that effect in the text, and have now emphasized and refined our phrasing throughout the text. We show theoretically that a system of point vortices should be hyperuniform, but following the simulation results we believe, as the reviewer noted, that a pure system of point vortices may never reach such a state, or may reach it at an infinite time. In other words, some form of perturbation that breaks the rotational symmetry of the Hamiltonian is required to reach hyperuniformity. We have tested a few forms of perturbations to the dynamics. In particular — numerical truncation errors and steric interactions. In both cases the system becomes hyperuniform. In the latter case the rate of convergence to a hyperuniform state does not depend on the stepsize. In the former case it does: as the timestep is reduced, longer times are required to reach hyperuniformity. We therefore believe, as the referee wrote, that a pure system with no numerical noise may never reach hyperuniformity. We have now tested both running a symplectic scheme and adding an additional form of noise as the referee suggested and outline the results below as well as in the text and the SI.

The referee asked about the errors in the conservation of \mathcal{H} and M at long times. For 10,000 point vortices, the Hamiltonian is conserved with a relative error of < 0.001 up to 16,000 cycles, the second moment is conserved up to a relative error of 4×10^{-4} . We find these errors to be satisfactory. We have now included in the Methods section details of the conservation of the second moment of vorticity as well. Our reasoning for choosing a Runge-Kutta method with an adaptive time step was two-fold:

(1) high-order accuracy meant that low truncation errors could be specified and simulations could be run for many cycles in a reasonable wall-time, and (2) since the Hamiltonian and second moments are not explicitly preserved, the conservation of these quantities can be used as a measure of numerical fidelity. In contrast, symplectic schemes, particularly when the Hamiltonian is non-separable (as is the case here), have several associated challenges: (1) to obtain higher than first-order accuracy, careful ordering of computations has to be preserved, making the simulations hard (if not impossible!) to parallelize (and non-amenable to fast algorithms, such as a fast-multipole method), and (2) since the conserved quantities are explicitly conserved, a basic marker of numerical fidelity is lost — meaning that while conservation looks great the actual dynamics could be very wrong.

In particular, the first issue (relating to parallelizability), means that these simulations run far slower: our Runge-Kutta based simulations were run on 128 core nodes using reasonably well optimized code that made effective use of all cores. We have now implemented and run a symplectic integration scheme (see below), but unfortunately, this code is serial (with no clear way to achieve a parallel implementation), and suffers from poor memory access patterns, and hence is slower by orders of magnitude (not to mention less accurate — second order with fixed timesteps, as opposed to fifth or eighth order for the RK schemes, with fixed or adaptive timesteps). We thus had to use a smaller ensemble (1000 particles). When measuring hyperuniformity in an open system, using a smaller system size is problematic. Finally, we cite Min, Mezic and Leonard, (Phys. of Fluids 1996) “Because of the chaotic nature of the vortex motion, error propagation during numerical computation is inevitable. This is a clear manifestation of the ‘sensitive dependence on initial conditions,’ and there is no way around it.”

Although we have not reported every test we have done in the manuscript, we have run many simulations that have explored the effect of initial data. We have typically run experiments using data generated on-the-fly (by rejection sampling a disk of radius R , as explained in the methods section of the paper). For fixed numerical and physical parameters, we have not observed any evidence that the specific realization of this kind of randomly generated data affects the final state of the system nor the timescale that it takes for the system to evolve to that state. We do note that in some cases (e.g. when trying to compare the effect of numerical parameters such as the error tolerance or timestepper), we have purposefully initialized simulations with exactly the same initial data, to eliminate a potential source of variation.

The referee made excellent recommendations which we have tested:

- *White noise*

We have added random white noise to the simulations. We tested the effect of noise for ensembles of 1,000 particles, with noise levels which are larger than the tolerance of the scheme/the truncation error. We compared results with and without noise by plotting the lowest $S(q)$ value as a function of time. This is a quick way to estimate the convergence to a steady state. For a noise level with a standard deviation of $D = 0.001$ or $D = 0.0001$, the rate of convergence to hyperuniformity did not change by the addition of noise. Taking a larger noise, with a standard deviation of $D = 0.01$, resulted in no apparent hyperuniformity. We expect that a large diffusion will destroy the conservation laws entirely. We note that this noise was selected completely at random, and was not correlated through the mobility tensor; efficient implementation of such a scheme is nontrivial (see e.g. Sokolov and Diamant, J. Chem. Phys., 2018).

- *Symplectic integration*

We have used the scheme developed by Zhang and Qin (*Computers. Math. Applic.*, 1993) which incorporates the analytical solution of two point vortices to construct a symplectic scheme. The time to compute single substages of the timestepper is *at least 100 times more costly than a non-symplectic scheme*, as it does not admit parallelization and has poor memory access patterns. We opted instead to test a smaller ensemble of 1,000 particles over $\sim 10^5$ cycles. We have tested timesteps of 10^{-6} for which the Hamiltonian is conserved with a relative error of 10^{-4} , and the second moment is conserved with a relative error of 10^{-11} . In comparison, using ODE853 with a tolerance of 10^{-6} for 1,000 particles, integrated over the same amount of time, results in conservation with a relative error of 2×10^{-3} in the Hamiltonian and 6×10^{-4} in the second moment. The symplectic runs showed little to no signs of hyperuniformity (see figures in the SI) — however, these are smaller ensembles simulated with a low-order, non-adaptive timestepper.

While the Hamiltonian and second-moment are well conserved, the dynamics are nevertheless not guaranteed to be accurate. We have also tested starting the simulation from a hyperuniform state and running it using the symplectic scheme for a long time ($\sim 5 \times 10^4$ cycles). In this case, hyperuniformity persisted. We present the radially averaged structure factor in the SI. Since the symplectic scheme very nearly preserves rotational symmetry of the system, these results reinforce our previous suspicion — that perturbations that break rotational symmetry are needed to overcome the dynamical bottleneck.

- *Other timesteppers*

In an effort to assess whether the *specific* choice of non-symplectic timestepper might affect the results, we re-implemented the point-vortex portion of our code in Julia, in order to access the large selection of timesteppers implemented in the DifferentialEquations.jl package. We tested several of the high-order adaptive methods, including the “Tsitouras-Papakostas 8/7 Runge-Kutta method” and the “Verner’s ‘Most Efficient’ 9/8 Runge-Kutta method”, as well as this packages implementation of the eighth-order Dormund-Prince scheme used for many of the simulations in the paper. For all timesteppers, the rate at which hyperuniformity appears depends on the size of the truncation error, and those rates are approximately consistent across methods.

Minor comments

- 1 **Referee:** In the subplots showing the 2D $S(q)$, I find the color scheme (black = zero) confusing. I recommend it is inverted.

Response: We understand the confusion and have tested the inverted color scheme. We decided to stay with the current scheme as the features are more observable to the naked eye. This scheme is also in line with typical diffraction scattering experimental plots.

- 2 **Referee:** Page 2, 11 lines from the end: typo – extra ‘(‘.

Response: corrected!

- 3 **Referee:** What happens at low densities? Is the steady state still hyperuniform?

Response: Indeed so it seems. In Fig. 3C we show results for various area fractions, ϕ . We have tested down to $\phi = 0.04$ and even at that area fraction we see the beginning of hyperuniformity. The lower the density the slower the dynamics is so we have not attempted densities lower than that.

- 4 **Referee:** The definition of returnity can be clarified.

Response: We agree with the referee and have tried to clarify the definition in the revised version. The returnity is now called “the relative deviation” and its meaning redefined in the text and in the caption of the figure.

- 5 **Referee:** Entropy measurements and Fig 4E: The changes in entropy are very small – about 1 – 2% with an error of 0.5 – 1%. This is not surprising given that converting the data into a figure with KxK pixels, estimating the entropy using lossless compression will be severely undersampled unless the number of realizations is of the order of 2^{K^2} . As the authors already calculated the structure factor, I suggest applying the method of [43]. Also, it is interesting to plot the entropy as a function of Γ_h/Γ_l .

Response: The referee is correct regarding the large errors in the compression plot. We did not attempt a rigorous calculation but merely aimed at showing that the simulations follow the theoretical reasoning — slow vortices expand thereby increasing entropy, fast vortices slightly compress and the entropy of the mixed system increases. Following the referee’s recommendation, we have considered applying the method of Ref. 43 (Ariel and Diamant PRE 2020), but it does not seem to be immediately applicable to a mixture. From Ref. 43: “A key disadvantage of entropy estimation based on density correlations is that it is more particular. While sampling, and especially compression, can be applied ‘blindly’ without prior knowledge of the system, relations such as Eq. (3) are limited to systems of a certain category. For example, Eq. (3) must be modified if it is to be applied to mixtures or to anisotropic particles.”

Report and response to Referee 3:

Referee: By considering Hamiltonian formalism, the authors predict that point vortices should form disordered hyperuniform states due to conservation laws. The authors also recover the exactly the same result for torque driven membrane rotors with a finite size, in the limit of short separations. The theoretical calculations are supported by detailed hydrodynamic simulations, with an excellent agreement between the two. Simulations are then used to consider a 50:50 mixtures of fast and slow spinning point vortices. Here, phase enrichment is observed for both population, while hyperuniformity is observed for fast spinning vortices only. The two populations occupy two different areas, which is shown to maximize the number of available states.

From the point of view of rotational active matter, I do find the presented results interesting. It is fascinating that the conservation laws stemming from the Hamiltonian structure of the rotationally invariant interactions result to the distinct states, as shown by the authors. Especially the phase enrichment in the mixtures is very nice. I am not a theorist nor too familiar with the formalism used, so some details may have escaped from me. The authors make an effort to simplify the mathematics, but at places I was lost what were the basic assumptions vs derived results.

Response: We thank the referee for stating the results are interesting and nicely presented! We also found it fascinating that there are distinct structural states stemming from the conservation laws. We agree with the reviewer that what we assumed, and what our derived results were, was not always clear in our original manuscript. We have made considerable changes — rewording and reorganization of text — and hope that the resulting revised manuscript is much clearer in this regard.

Let us also summarize here as well what is assumed and what was derived. We start with the equations for vortices in an inviscid fluid. It is known that one can write three conservation laws for point vortices — the Hamiltonian, and the first and second moments of vorticity. We write in Eq. 2 the Stokes equations for membrane rotors and show that the dynamics are exactly the same as for vortices in the limit of $r \ll \lambda$, where λ is the Saffman-Delbrück length. For rotors, we assume that the density is low such that the main hydrodynamic contribution of a rotor to the flow is of a point torque. We mainly focus on the limit of small distances relative to the Saffman-Delbrück length, but comment on the other limit in the text (e.g. Fig.1B and 2B and C) and in the SI. It is known in Hamiltonian mechanics that symmetries of the Hamiltonian correspond to conservation laws. In our case, the conjugate variables of the Hamiltonian are the actual locations of the particles. Thus, the Hamiltonian is geometric in nature and conservation laws associated with it limit the distribution of the particles. To get a sense of the meaning of the conservation laws:

- The Hamiltonian is analogous to the kinetic energy (for point vortices in an ideal fluid it is, in fact, the actual kinetic energy). Since \mathcal{H} is a sum of $-\log$ of the distances between each two vortices, its conservation means that two vortices cannot overlap or be too close since then the Hamiltonian will diverge.
- The second moment of vorticity is also conserved which can be shown from invariance to rotations. The second moment is analogous to the moment of inertia, except that instead of masses the sum over distances squared is weighted by the circulation. Since the second moment is conserved, two particles cannot scatter to large distances, as then the second moment will diverge.
- The first moment of vorticity is analogous to the center of mass. Again, instead of masses the sum is weighted by circulations. Its conservation corresponds to a fixed center of vorticity.

We thus see that particles cannot be too close to each other nor too far apart, which hints that the system should be hyperuniform. Mind you, hyperuniformity is the suppression of density fluctuations. We go on to show this should be the case: we derive a link between the Hamiltonian and the structure factor, Eq. 1. This integral relation puts a bound on the structure factor. For point vortices, we find that at small wavenumbers $S(q) \propto q^\alpha$ with $\alpha > 0$ which is one of the definitions of a hyperuniform material.

We have revised the manuscript to include the key-points mentioned above.

Other comments/questions:

- 1 **Referee:** On the page 3, where the steady state is characterized by 3-distinct ways. The role of perturbations is discussed. It is stated that some amount of perturbations are needed to observe hyperuniformity. While this would not be a problem in an experimental setting, does it imply that if the Hamiltonian and its moments are exactly conserved, hyperuniformity would not be observed for point vortices?

Response: Yes, we believe the referee is correct. As we show theoretically, a system of point vortices should be hyperuniform from the conservation laws. However, in simulations of point vortices without steric interactions, taking smaller and smaller step sizes results in hyperuniformity emerging at later and later times. It is impossible (or very hard) to show what will happen in a pure system with zero numerical errors, but we suspect that hyperuniformity will not be reached, or perhaps, it will take an infinite time to reach the hyperuniform state. It seems that perturbations that break rotational symmetry are needed. As the referee points out, in any experimental system this will not be a problem. We have now also tested a symplectic integration scheme which by design conserves the rotational symmetry of the Hamiltonian, and do not observe a clear sign of hyperuniformity in it (however, since the scheme is much more computationally expensive, we were only able to test a lower order timestepper and a smaller system of 1,000 particles). We mention the new simulation and its results in the main text and the SI, and clarify the role of perturbations in the main text.

- 2 **Referee:** The $S(q)$ for rotors (inset in Fig. 3B) shows plateau at small q of all $\phi > 0.14$. While in the main text it is said that all the samples showed hyperuniformity. This seems contradictory with the definition of $S(q) \rightarrow 0$ when $q \rightarrow 0$. Also, the rotors probably develop (local) hexatic order before the crystallization observed at $\phi \sim 0.5$. It would instructive to plot the local hexatic order parameter as a function of ϕ .

Response: Indeed the referee is correct in that when steric interactions are added local order emerges. We have tested smaller systems of 2,000 particles for the plot in Fig. 3B. The ensemble transitions from a disordered hyperuniform state at low densities to an ordered hyperuniform state in the form of a hexagonal lattice at high ones. In future work we plan to investigate this transition in depth. A plateau in the structure factor does not mean that there is no hyperuniformity. The structure factor of a pure lattice, for example, is what is called ordered hyperuniform. And, for a lattice, the structure factor is zero except for sharp delta peaks. This is what we start to see for the highest concentration $\phi \sim 0.54$. For a finite system, the value of the plateau will not be zero but some finite, small, value. The referee made a good suggestion to measure the local hexatic order parameter, which we have now added to the text and the figure.

- 3 **Referee:** I was somewhat confused with the returnity. The returnity seems to be close to zero at longtimes. Logically to me this suggests that the particle position has deviated from its original position. However, the definition of the returnity in the caption suggests that zero returnity means no deviation. The calculation/discussion should be clarified.

Response: The referee is correct that the definition was confusing and misleading. We have renamed the returnity and now simply call it the relative deviation. If a particle has returned to its previous position its value will be zero, and its max value is given when the particle has traveled the entire perimeter, whence its value is 2π .

- 4 **Referee:** Connected to above, from the main text it seems that the system is not in absorbing state at long times, while it is also said that the ensemble undergoes a solid body rotation, where the particle i returns to its position cyclically. This paragraph should be clarified.

Response: The referee is again correct that the writing was confusing. We have changed the manuscript accordingly. The average velocity increases linearly with radius and in that sense we meant it is like a solid body rotation. The system does not appear to reach an absorbing state. Tracking a single particle over long times shows that it still explores the entire phase-space (which

in our case is equivalent to configurational-space). The relative deviation (previously named “the returnity”) only measures the deviation from the previous position after *one cycle*. This deviation is initially large on average but is small when the system is at steady state. We now added a clarification to the text as well as a plot of the trajectory of a few particles over many cycles (~ 130). These trajectories show Brownian-like dynamics which will be interesting to further explore, as we hope to do in future work.

- 5 **Referee:** For which system is the returnity measured? Would it be expected to be different between point vortices and rotors?

Response: We measured the returnity for a system of point vortices. Following the referee’s comment we have now measured the returnity at a low area fraction $\phi = 0.1$ of rotors as well and have gotten a similar plot to Fig. 3D. For rotors, as the concentration is increased, the ensemble crystallizes. Therefore, at higher area fractions the system does reach an absorbing state. We have added a clarification to the text.

- 6 **Referee:** The role of fig. 2c is not clear to me. Does this mean that at long separations (low area/volume fractions) the finite size rotors are not expected to have similar steady state as predicted for point vortices?

Response: Indeed there is a difference between cases where the total size of the ensemble is smaller than the Saffman Delbruck length which is a few microns, to cases where the ensemble size is larger than λ . For a case where the density is low and the number of particles is large, the scaling of the interaction between faraway rotors is $1/r^2$. That is, lower than the $1/r$ for interactions between close rotors. The symmetries of the Hamiltonian are the same and there are still conservation laws bounding the distribution of particles. However, Eq. 1 gives a lower bound of $S(q) \sim q^\alpha$ with $\alpha > -1$ from which we cannot conclude the system should be hyperuniform. Simulation results in this limit indicate that the system is hyperuniform, as we show in the Supplementary Information.

- 7 **Referee:** Also, if not completely obvious, it would be nice if arguments/simple explanations were given for example for the symmetries for a translation in time, space and rotation corresponding to the conservation laws. This would increase the accessibility of the article. Can these be understood in some simple way, such as conservation of the hamiltonian and 2nd moment kind of fixes the density, while the vanishing first moment corresponds to no center of mass movement? At longtimes this then leads to a steady state with a (close to) solid body rotation, as opposite to the chaotic dynamics at times near the random starting configuration.

Response: Yes! Exactly as the referee wrote. The conservation of the Hamiltonian and of the 2nd moment of vorticity limit the distribution of particles. The second moment of vorticity is analogous to the conservation of the center of mass. We have now included these comments in the text. We have added further details on the conservation laws in response to the first comment by the referee.

- 8 **Referee:** In the ref 15, for torque driven rotors it is suggested that the steady state is characterized by extrema of a ”pseudo kinetic energy”. In the formalism presented here, is there an analog to this?

Response: Yes, the case of torque driven rotors in Ref. 15 is the same as our case, and so here as well, the Hamiltonian could be said to be a pseudo kinetic energy. In Ref. 15 they claim that an hexagonal lattice arises due to minimization of this energy. However, they then show that the hexagonal state is only marginally stable. This is a result of the fact that interactions between the rotors are not repulsive nor attractive. For example, two rotors will orbit around each other at a fixed distance, they will not draw nearer or separate. We show in Ref. 17 that adding steric interactions, this marginally stable state can become stable. The same happens here with steric interactions at sufficiently high concentrations. Due to the marginal stability, we do not expect nor do we see spontaneous crystallization in a system of point vortices.

9 **Referee:** The ψ in the Hamiltonian above the equation 5 on page 3 is not defined.

Response: The referee is correct; this was a typo and now reads Ψ .

REVIEWERS' COMMENTS

Reviewer #2 (Remarks to the Author):

Second report on "Hyperuniformity and phase enrichment in vortex and rotor assemblies" by Oppenheimer et al. The authors addressed all my comments. This is an excellent paper. I recommend it is accepted to Nature Communications.

I have two minor comments:

1. This is a matter of style, but I think the abstract can be improved. The revised version is detailed and specialized, which somewhat obscures the main results.

2. It is worth noting that the observed divergence of the relaxation time towards hyperuniformity is consistent with the critical slowing down reported, for example, in [41]. Interestingly, in [41], hyperuniformity is only obtained at critical values of the parameters. Here, it does not require fine tuning. Instead, numerical errors may play an analogous role.

Reviewer #3 (Remarks to the Author):

I would like to thank the authors for the detailed and comprehensive replies. These fully address my queries. Only very minor comment regarding the supplementary fig. 4, which the y-label reads returnity.

I feel that the authors have considerably improved the manuscript, and I am happy to recommend publication in Nature Communications.

We were very pleased that both referees recommended the paper for publication. As we stated previously, we feel that not only has the manuscript improved by addressing their comments, but also our understanding of the system. The reviewers had additional minor comments which we address below and in the revised text. We also made revisions according to the editorial requests. Below we copy the referee reports in green. Our response is in black below each point.

Report and response to Referee 2:

Referee: Second report on “Hyperuniformity and phase enrichment in vortex and rotor assemblies” by Oppenheimer et al. The authors addressed all my comments. This is an excellent paper. I recommend it is accepted to Nature Communications.

Referee: I have two minor comments:

- 1 This is a matter of style, but I think the abstract can be improved. The revised version is detailed and specialized, which somewhat obscures the main results.

Response: We agree with the referee that the new version of the abstract is a bit too specialized. We have changed the abstract back to what it previously was, with much smaller corrections.

- 2 It is worth noting that the observed divergence of the relaxation time towards hyperuniformity is consistent with the critical slowing down reported, for example, in [41]. Interestingly, in [41], hyperuniformity is only obtained at critical values of the parameters. Here, it does not require fine tuning. Instead, numerical errors may play an analogous role.

Response: We thank the referee for pointing out that our observations are consistent with the critical slowing down towards hyperuniformity in the absorbing states of e.g. [41]. We have now included such a statement:

“The observed relaxation towards hyperuniformity is consistent with the critical slowing down reported for other systems (e.g. [41])”

Report and response to Referee 3:

Referee: I would like to thank the authors for the detailed and comprehensive replies. These fully address my queries. Only very minor comment regarding the supplementary fig. 4, which the y-label reads returnity.

I feel that the authors have considerably improved the manuscript, and I am happy to recommend publication in Nature Communication.

Response: We thank the referee for recommending the paper for publication in Nature Communications.

We have now re-labeled Fig. 4 in the Supplementary Information, and call it the Relative Deviation.